# A Unified Multimodal Interface for the RELAX High-Payload Collaborative Robot

**DOI:** 10.3390/s23187735

**Published:** 2023-09-07

**Authors:** Luca Muratore, Arturo Laurenzi, Alessio De Luca, Liana Bertoni, Davide Torielli, Lorenzo Baccelliere, Edoardo Del Bianco, Nikos G. Tsagarakis

**Affiliations:** 1Humanoids and Human-Centered Mechatronics Research Line, Istituto Italiano di Tecnologia (IIT), Via Morego 30, 16163 Genova, Italy; 2Department of Informatics, Bioengineering, Robotics and Systems Engineering (DIBRIS), Università di Genova, 16145 Genova, Italy; 3Department of Information Engineering (DII), Università di Pisa, 56122 Pisa, Italy; 4DISI, Università di Trento, 38123 Trento, Italy

**Keywords:** human–robot collaboration, physical human–robot interaction, multimodal interface for HRI, high payload cobot

## Abstract

This manuscript introduces a mobile cobot equipped with a custom-designed high payload arm called RELAX combined with a novel unified multimodal interface that facilitates Human–Robot Collaboration (HRC) tasks requiring high-level interaction forces on a real-world scale. The proposed multimodal framework is capable of combining physical interaction, Ultra Wide-Band (UWB) radio sensing, a Graphical User Interface (GUI), verbal control, and gesture interfaces, combining the benefits of all these different modalities and allowing humans to accurately and efficiently command the RELAX mobile cobot and collaborate with it. The effectiveness of the multimodal interface is evaluated in scenarios where the operator guides RELAX to reach designated locations in the environment while avoiding obstacles and performing high-payload transportation tasks, again in a collaborative fashion. The results demonstrate that a human co-worker can productively complete complex missions and command the RELAX mobile cobot using the proposed multimodal interaction framework.

## 1. Introduction

The use of collaborative robots in industrial settings has recently gathered great attention as a means of transitioning from traditional industrial robotic systems that operate in separated and enclosed workspaces to robots that can provide flexibility while operating safely alongside their human coworkers. This has led to the development of various collaborative robots, known as cobots [1], which are intended to be used in industrial tasks, leading to increased productivity and to reduced operating costs and downtimes [2].

Various research laboratories and companies have expended effort on integrating cobots and mobile robots to realize mobile cobots, allowing the execution of tasks that require both mobility and manipulation functionalities. A distinctive feature of this kind of robot is that they provide physical assistance in terms of force in collaborative tasks while simultaneously allowing safety when working with their human counterparts in the whole industrial/warehouse environment [3]. Application domains such as logistics operations, robot companions, and co-workers in manufacturing and construction spaces can benefit from effective human–robot cooperation with mobile cobots. In particular, tasks with high physical requirements involving robots assisting human workers in transportation of heavy loads and navigation in industrial/warehouse environments are receiving increasing attention. These scenarios require the mobile collaborative robot to actively read the intention of human coworkers and react accordingly [4]. For the above, an embedded multimodal interface allows the robotic system to adapt to the movement intentions of the human operator [5]; this communication can occur using more than one modality, and can be accomplished intuitively and efficiently to best suit the individual situation [6].

One of the main motivations for using a multimodal interface to control a mobile cobot is to improve the ease of use and user experience for the operator. By providing multiple input modalities, the operator can choose the one that is most comfortable or appropriate for the task at hand. This can help to reduce the cognitive load on the operator, as they do not have to remember complex commands or manipulate small buttons and controls. Additionally, multimodal interfaces can improve the safety of the cobot, as they allow the operator to retain their attention and stay focused on their tasks. This can be especially important in industrial or hazardous environments where the operator needs to be able to quickly and easily control the cobot to avoid potential accidents or collisions. Finally, by providing multiple input modalities, multimodal interfaces can improve the flexibility and adaptability of the cobot, as it can be used in a wider range of applications and environments.

In the work presented herein, we propose a novel unified multimodal and multisensor interface framework for a new mobile co-worker platform called RELAX (Robot Enabler for Load Assistive relaXation). RELAX allows for the control of both mobility and manipulation by integrating a physical interaction interface, an Ultra Wide-Band Real-Time Location System (UWB RTLS), a mobile GUI, and finally verbal and gesture recognition interfaces. Specifically, physical interaction is achieved without the need for any additional mechanical/electronic interface by utilizing the compliance and force/torque sensing capabilities of the RELAX cobot. In addition, UWB technology is used for mobility control; a set of anchors is placed on the robot and a tag device is worn by the user, as described in Section 4.2, allowing the operator’s pose to be estimated relative to the cobot. Thus, it can be used in crowded environments without the need for perception sensors, which may be unreliable. The proposed multimodal interface framework is demonstrated on the RELAX mobile cobot, which represents a contribution of this work. In fact, the RELAX platform has the physical capabilities to be used in realistic manufacturing tasks involving high-payload conditions. Specifically, the RELAX cobot has the ability to mimic human strength and can carry heavy loads; at the same time, it is equipped with sensors that can detect the force being applied to it, allowing it to handle tasks that require high interaction forces while maintaining safety and adaptability during physical interactions. These payload and interaction sensing capabilities are combined with a large workspace that makes the RELAX platform suitable for a variety of tasks, while its overall volume and dimensions permit its integration into existing manufacturing workstations.

We have validated the proposed framework on a set of navigation/transportation collaborative tasks using the recently developed RELAX mobile cobot, showing the effectiveness of our multimodal interface and demonstrating its advantages compared to a single-channel control interface.

The rest of the paper is organized as follows: Section 2 describes the related works, Section 3 illustrates the RELAX cobot, Section 4 introduces the multimodal interface framework to control it, and Section 5 presents the experimental trials and discusses the results for the proposed approach. Section 6 then presents our conclusions and a brief discussion of future work.

## 2. Related Works

In the field of colborative robotics, ABB introduced the dual-arm YuMi cobot (https://new.abb.com/products/robotics/collaborative-robots/yumi/irb-14000-yumi), which consists of two 7-DOF arm systems designed for small-part assembly side-by-side with human workers. Even though YuMi can detect contacts, it is position-controlled [7], offering minimal functionality as a cobot. The torque-controlled KUKA LBR IIWA (https://www.kuka.com/en-gb/products/robotics-systems/industrial-robots/lbr-iiwa) is the new generation of the LBR (7-DOFs arm), which has been widely used in robotics research [8] as well as industry [9]. Franka Emika has recently introduced two 7-DOF arms named Production 3 (https://www.franka.de/production/) and Research 3 (https://www.franka.de/research/) designed for assembly and component testing side-by-side with human workers. Both models are fully torque-controlled and have a moderate payload (3 kg), a small footprint, and high repeatability. Universal Robots has developed a set of collaborative robots (https://www.universal-robots.com/products/) called UR3e, UR5e, UR10e, UR16e, and UR20 for use in manufacturing and assembly environments. They feature lightweight designs and 6-axis arms that can be programmed to perform a wide range of tasks while working with payloads of 3 kg, 5 kg, 10 kg, 16 kg, and 20 kg, respectively. The Japanese company Fanuc has several collaborative robot solutions on the market; for example, the CR-14iA/L cobot (https://www.fanuc.eu/it/en/robots/robot-filter-page/collaborative-robots/collaborative-cr-14ial) has a payload of 14 kg and a medium footprint. It can be used for loading and unloading machines as well as for other factory automation tasks. In the research community, the Fetch Robotics Fetch mobile manipulator (https://fetchrobotics.com/fetch-mobile-manipulator/) was designed for use in manufacturing, logistics, and research environments. It features an autonomous navigation system, can be programmed to move around a facility and to pick up and transport items as directed, and has an arm payload of 6 kg.

The above-described robotic arms demonstrate relatively low payload-to-weight ratio performance, which restricts the physical assistance and payload that these cobots can support. In [10], the authors reported an extensive analysis of the available commercial cobots. In particular, the weight was compared with the payload, showing a very limited payload-to-weight ratio. Figure 1, taken from [10], shows the correlation between weight and payload among most of the currently available cobots.

In this work, we focus on a multisensor direct interface for one-way communication from the human to the robot. The proposed interface system works by monitoring, classifying, and converting a set of multimodal signals from the user into appropriate outputs for mobility and manipulation of the cobot in a closed-loop fashion. Most of the literature regarding the above topics is focused on simple joystick interfaces [11] and Graphical User Interfaces (GUIs) [12], which are not effective for industrial/warehouse environments. Similarly, verbal and visual interfaces for speech and gesture recognition systems [13] have limited control over the navigation of the mobile cobot. Haptic devices providing force feedback [14] usually have only six DOF control, i.e., they can control only a fixed manipulator. Finally, physical human–robot interaction (pHRI) methodologies have been adopted, such as in [15], where the MOCA functions autonomously while the operator can choose whether to be physically coupled with it through a mechanical admittance interface. Similarly, teleoperation interfaces based on remote control of the robot using 3D motion tracking devices and electromyography (EMG) data [16,17] have been adopted, especially in the research community.

## 3. The RELAX Mobile Cobot

The RELAX mobile manipulation collaborative robot represents a customized system composed of a commercial mobile industrial platform, the non-holonomic MIR250 (https://www.mobile-industrial-robots.com/solutions/robots/mir250/), into which a custom collaborative robotic arm has been integrated.

The motivation for this choice was based on the objective of delivering a high-payload collaborative robot arm capable of executing task scenarios imposing high payload conditions or demanding tasks with high-level interaction forces on real-world scales, such as carrying and transporting heavy payloads in warehouses, using heavy tools, and wall drilling or cutting operations, e.g., for electrical installation activities on construction sites.

In particular, the body of the RELAX robot arm is composed of a number of 2-DOF body modules that integrate two actuator sizes developed to achieve a continuous payload capacity of 14 kg.

To implement the kinematics of the RELAX collaborative robot arm, two types of structural cell modules, an L-shape and a T-shape, were designed, each of them housing two actuation units with their corresponding motor drive electronics to form 2-DOF modular sub-bodies Figure 2. The active length of the RELAX robot arm is 1.0 m, while the overall length of the robot from the base to the robot end-effector flange is 1.3 m. The total weight of the RELAX robot arm is 23 kg.

### 3.1. RELAX Actuation System

The actuation unit of the RELAX arm is implemented using the foundation actuation technology in [18], and consists of a DC brushless motor combined with a harmonic drive reduction unit. Motor, gear, and joint sensing have been highly integrated to optimize the weight and dimensions of the joint, favoring a high payload density on the part of the overall robot.

The sensing system of the actuator includes two position sensors, one torque sensor, and two temperature sensors. One encoder monitors the position of the motor side shaft, while another encoder measures the position of the output link side shaft pulley (harmonic drive output). A torque sensor is based on a custom load cell structure sized appropriately for the two sizes of the actuator to measure the torque generated at the output. The actuation unit includes a brake module. Finally, the actuator incorporates two temperature sensors, which are installed to monitor the motor windings and motor electronics temperatures. The main specifications of the two actuator sizes of the RELAX robot are displayed in Figure 3.

### 3.2. RELAX Arm Prototype

The fabrication of the RELAX custom collaborative robot arm was also performed starting from the manufacturing, assembly, and calibration of its actuators followed up by the realization of the 2-DOF body modules (Figure 4) and the final assembly of the full RELAX collaborative robot with the integration of the mobile platform (Figure 5).

The final integration of the RELAX robot arm on top of the MIR250 is reported in Figure 6, and its technical specifications are summarized in the table in Figure 7.

### 3.3. RELAX End-Effector

To render the RELAX arm capable of grasping and holding objects and tools of various shapes and sizes with low to moderate payload and interaction force requirements, a robust and easily reconfigurable end-effector module was fabricated. It consists of a generic 1-DoF jaw-type gripper (shown in Figure 8) in which the bottom jaw is fixed while the top jaw is powered by a single actuator housed inside the gripper’s main body. Both the bottom and top jaw are designed and integrated in a modular way, allowing for fast replacement with jaws using appropriate materials, stiffness, and geometry to fulfill specific applications. The torque-controlled actuator delivers a pinching force at the tip higher than 50 N. The gripper is equipped with the same electromechanical interface integrated into the RELAX wrist flange, making for fast mounting and integration into the RELAX arm system.

### 3.4. RELAX Software Architecture

As shown in Figure 9, the cobot’s two main subsystems (MIR250 and the collaborative arm) are able to communicate thanks to a ROS Bridge Client/Server (http://wiki.ros.org/rosbridge_suite) that translates the Web API provided by the MIR250 to the ROS communication language. On the arm side, the XBotCore [19,20] real-time software framework assures the abstraction of the two hardware subsystems, allowing for real-time control of the EtherCAT network of the arm and end-effector as well as whole-body control of the integrated mobile cobot using the CartesI/O engine [21], with built-in ROS support enabling fast integration of GUIs and extra hardware devices. More specifically, the ROS infrastructure is mainly used for remote inter-process communication, including the publish–subscribe paradigm and service calls. Moreover, we employed the ROS1 Navigation stack (http://wiki.ros.org/navigation) package on the mobile base for 2D planning. Using the odometry and sensors data retrieved from the MIR250 ROS Bridge as input, the mobile cobot can be driven to a certain goal pose on the map using the velocity commands generated by the navigation stack.

The RELAX software stack includes simulation support for fast prototyping and transparent code porting to the real robot; built-in support is provided for the Gazebo dynamic simulator, as shown in Figure 10.

## 4. Multimodal–Multisensor Interface Framework

The diagram in Figure 11 shows an overview of the proposed unified multimodal–multisensor interface framework for mobile cobots, with their application on the RELAX cobot on the right. On the left of the image, the physical devices/sensors used to control the robot are depicted, while the center shows the five interfaces, composing the multimodal framework plus their handler. In the following sections, we describe each of the components making up this framework in detail.

### 4.1. Physical Interface

The proposed physical interface permits control of the mobile base velocities interacting physically with the robot arm. By rendering low joint impedance settings on the cobot, the Cartesian displacement of the end-effector in the X and Y directions with respect to its starting pose generated by the interaction of the operator with the cobot is used to compute velocity references for the RELAX mobile platform. Moreover, the low impedance settings assure user safety during the physical contact between the operator and the cobot.

It is worth noticing that the physical interaction of the user can happen in every part of the robot body, making it very intuitive and straightforward to use. This perturbation generates translational and rotational velocity references for the mobile base following the linear controller described in Equation (Equation 1).
(1)$$\begin{array}{c} v_{x}\left(t\right)=k_{v}\;\left({{}^{\mathrm{base}}EE}_{startX}-{{}^{\mathrm{base}}EE}_{currentX}\left(t\right)\right)\\ \omega \left(t\right)=k_{\omega }\;\left({{}^{\mathrm{base}}EE}_{startY}-{{}^{\mathrm{base}}EE}_{currentY}\left(t\right)\right) \end{array}$$
where $v_{x}\left(t\right)\in R$ represents the translational velocity reference for the mobile base, ω(t)∈R represents the rotational velocity reference for the mobile base, ${{}^{\mathrm{base}}EE}_{startX}\in R$ and ${{}^{\mathrm{base}}EE}_{startY}\in R$ denote respectively the arm end-effector X and Y position w.r.t. the cobot’s base calculated when the physical interface is started, and ${{}^{\mathrm{base}}EE}_{currentX}\left(t\right)\in R$ and ${{}^{\mathrm{base}}EE}_{currentY}\left(t\right)\in R$ represent respectively the current arm end-effector X and Y position w.r.t. the cobot’s base, and $k_{v}\in R$ and $k_{\omega }\in R$ are the proportional gains respectively for the translational and rotational part. In Figure 12, a use-case of the above-described physical interface can be found with the operator generating a forward translational velocity reference at t_1, and a backward translational velocity reference plus an anti-clockwise rotational velocity at t_2 for the mobile base.

### 4.2. UWB Interface

Ultra Wide-Band (UWB) is a type of Radio Frequency (RF) technology characterized by a large bandwidth and low energy level transmission protocol, features that make this technology suitable for ranging and positioning systems. As highlighted in yellow in Figure 11, the input for the UWB Interface is the TREK1000 system manufactured by DecaWave [22], which provides configurable units that can be used as anchors (fixed elements) and tags (moving elements) for applications such as tracking or geo-fencing. In the proposed framework, the cobot’s base is equipped with four anchors placed in known positions, and the operator wears a tag. Thanks to the true-range multiliteration method [23], we are able to estimate the 3D position of the operator with respect to the reference system defined by the known positions of the four anchors. Moreover, a set of second-order filters is employed to process the input signals from the TREK 1000 system. Based on the above, the UWB interface is able to generate the cobot’s base velocity references to track the human pose with respect to the cobot mobile base at a requested relative distance and angle specified by the user. Figure 13 introduces a representation of the proposed UWB interface scenario.

The implemented controller for the mobile base is reported in Equation (Equation 2). The pose of the Tag (i.e., the operator) with respect to the base can be expressed as a function of the reference velocities for the mobile base, as in [24]; enforcing a linear controller, we can obtain
(2)$$\begin{array}{c} v_{x}\left(t\right)=\omega \left(t\right)\;{{}^{\mathrm{base}}Tag}_{Y}\left(t\right)\;-k_{v}^{uwb}\;\left({{}^{\mathrm{base}}Tag}_{dX}-{{}^{\mathrm{base}}Tag}_{X}\left(t\right)\right)\\ \omega \left(t\right)=-\frac{k_{\omega }^{uwb}\;\left({{}^{\mathrm{base}}Tag}_{dY}-{{}^{\mathrm{base}}Tag}_{Y}\left(t\right)\right)}{{{}^{\mathrm{base}}Tag}_{X}\left(t\right)} \end{array}$$
where ${{}^{\mathrm{base}}Tag}_{X}\left(t\right)\in R$ and ${{}^{\mathrm{base}}Tag}_{Y}\left(t\right)\in R$ represent the X and Y positions of the current estimated tag (i.e., the operator) with respect to the cobot’s base, ${{}^{\mathrm{base}}Tag}_{dX}\left(t\right)\in R$ and ${{}^{\mathrm{base}}Tag}_{dY}\left(t\right)\in R$ represent X and Y positions of the desired tag (operator) with respect to the cobot’s base, and $k_{v}^{uwb}\in R$ and $k_{\omega }^{uwb}\in R$ are the respective proportional gains of the translational and rotational parts.

### 4.3. Joypad Interface

The joypad interface realizes a velocity controller for the mobile base, and is capable of commanding both its linear and angular velocities. It is implemented using a standard twist command for the base, which is generated by a virtual joystick implemented as a built-in feature in the *ROS-Mobile* Android application [25].

### 4.4. Verbal Interface

As in the previously described joypad interface, the ROS-Mobile Android app was used to integrate a speech recognition module as a verbal interface inside the proposed multimodal framework, as reported in Figure 14.

In particular, we implemented the “Speech” widget inside the app using the native Android Java API in order to have access to the microphone of the mobile device (or a paired Bluetooth device), and used the Android speech recognition engine (https://developer.android.com/reference/android/speech/SpeechRecognizer) to provide a particular command associated with a word to the cobot as an open/close primitive for the end-effector or a Cartesian reference pose for the arm.

### 4.5. Gesture Interface

The gesture interface relies on a 2D RGB image input from a camera to detect the human hand keypoints and the normalized hand aperture as a distance between the mean point of the hand tips and the mean wrist and thumb base point divided by the palm length. This interface is built on top of Mediapipe [26], a high-fidelity hand and finger tracking solution. It employs machine learning (ML) to infer 21 different 3D landmarks of the hand. By exploiting these, a set of preplanned Cartesian motions for the arm can be commanded by the human, for instance, to perform manipulation tasks associated with a specific gesture.

### 4.6. Multimodal–Multisensor Interface Handler

This component is employed to obtain the unified multimodal interface framework. As can be seen in Figure 11, this layer of the architecture takes inputs from all the above-described interfaces, and is capable of generating an output for the mobile cobot in terms of arm, end-effector, and base control.

The logic behind this module is based on the estimated pose of the operator; the main goal is to combine the advantages of the presented interfaces in order to achieve more effective human–robot cooperation. Considering the features reported in Table 1, it can be seen that each of the presented interfaces has certain benefits compared to the others while at the same time presenting drawbacks.

For example, while the physical interface allows the user to drive and place the robot in a certain location with high accuracy, it requires the operator to always be in contact with the robot within a small interaction range, which can be stressful for the operator over the long term. On the contrary, the UWB interface does not have high accuracy, and its input signal can be very noisy; however, it permits control of the robot from a moderate distance even in crowded environments, and can be used either indoors or outdoors, as it is based on RF distance measurements. In this context, the joypad interface as implemented inside a mobile phone/tablet app can be used to move the cobot’s base with moderate accuracy even from long distances; however, the operator needs to hold the device and always provide velocity input using one hand. In the same way, the verbal interface can be used through the same app to send high-level commands to the cobot, such as opening or closing the end-effector; the accuracy of this interface is categorized medium, as in noisy environments the speech commands may not be recognized correctly. The gesture interface has the lowest accuracy, as the visual mode depends on the camera and lighting conditions. However, it can be used far from the robot, which is essential during high-payload tasks that can be dangerous for human coworkers.

In light of the above, the proposed unified multimodal–multisensor interface handler is capable of combining the physical and UWB interfaces for the mobility part based on the estimated operator location. In this way, the UWB interface can be used to avoid direct contact with the robot, reducing operator fatigue when there is no need for high accuracy. When the mobile cobot needs to be driven with high precision, e.g., through narrow passages, or has to navigate cluttered environments, the physical interface can be employed. At the same time, in certain situations the operator can bypass the hybrid physical or UWB mobility mode altogether and teleoperate the cobot’s base using the joypad interface. Finally, the mobile cobot can be operated with the verbal or gesture interface in circumstances when the base is not in motion; this is done for safety reasons, as it prevents sending an unwanted manipulation action when the operator is close to the cobot.

## 5. Experimental Validation

Three sets of experiments were carried out to validate the performance of the RELAX cobot and the proposed multimodal framework.

### 5.1. RELAX High Payload and Repeatability Validation


**Experimental Setup**


The first set of experiments focused on the RELAX collaborative arm. To prove its high-payload capacity and repeatability, we carried out a set of stress tests on the robot by holding and executing a motion trajectory with a 12 kg payload. In terms of the parameters employed during this task, in the payload test we explored 80% of the maximum reach of the RELAX arm for a 30 min experiment, while for the repeatability test we used a line segment trajectory along the Y axis with a starting location A placed at (0.43 m, −0.4 m, 0.54 m) and a final goal location B placed at (0.43 m, +0.4 m, 0.54 m) with respect to the arm base, resulting in a line segment trajectory of 0.8m along the Y direction of the arm workspace.


**Results**


The top plot in Figure 15 reports the temperature evolution of the joints over time. As can be noted, the temperature increase in all joints at the end of the experiment is below $3{\;}^{{}^{\circ}}$C with respect to the starting temperatures. Furthermore, an accuracy and repeatability test under the same payload condition was executed with the cobot moving at 0.2 m/s in a cyclic Cartesian motion task; the results are shown in the bottom plot in Figure 15, highlighting the accuracy and repeatability performance of the RELAX cobot.

### 5.2. Multimodal Interface Framework for High-Payload Transportation


**Experimental Setup**


The second set of tests sought to validate the proposed unified multimodal interface in the experimental scenario illustrated in Figure 16. In this experiment, the RELAX mobile cobot and a human co-worker need to avoid obstacles and arrive at three marked locations in the environment by following a particular order (i.e., first A, then B, then C). A manipulation task is performed at each marked location. In terms of parameters used in the experiment, the maximum linear velocity allowed for the mobile base was set to ±0.3 m/s and the maximum angular velocity to ±0.4 rad/s. The RELAX arm was controlled in low impedance mode to ensure safe and user-friendly control.


**Results**


In Figure 17, the results for the physical interface show the commanded translational and angular velocities for the base, which are strictly linked with the respective calculated Cartesian errors of the end-effector in the X and Y directions with respect to its starting pose, as described in Section 4.1.

Figure 18 displays the data recorded for the validation of the UWB interface. In this experiment, the requested relative distance between the operator and the center of the base of the cobot was −1.35 m, i.e., the operator commanded the robot from behind, with the relative angle being equal to 0. The results shown here are related to the global X and Y pose of the mobile base and of the operator calculated using the odometry of the MIR250. As can be observed with the reference tracking X (the dotted line), as described in Section 4.2, the UWB interface is capable of tracking the reference pose of the mobile base even if the UWB signal is noisy and contains outliers.

The gesture interface is employed for manipulation tasks; in its current state it supports four different hand gestures, which are shown in Figure 19 and depend on the estimated hand aperture. From left to right, the figure demonstrates the *close*, *fist*, *half-open*, and *full-open* gestures with the related estimated hand aperture percentages. Safe thresholds are employed to differentiate the four gestures, which are then coupled with specific manipulation primitives for the robot.

Finally, the unified multimodal–multisensor interface framework results are illustrated in Figure 20. In this case, the handler described in Section 4.6 is capable of automatically switching from the physical interface to the UWB interface and vice versa, allowing the advantages of both interfaces to be exploited. While the reported data are similar to Figure 18, it can be noted that the UWB controller is active only in the white areas of the plot; in the light blue highlighted area, the operator’s estimated pose is closer to the cobot, and the physical interface is activated, speeding up those phases of the navigation in which the cobot requires more precision in less time.

The effectiveness of the proposed multimodal interface framework is highlighted in Table 2, where the above-described navigation scenario is taken into account.

The multimodal approach was compared with the standalone physical and UWB approaches, repeating the experiment five times for each interface. Three main metrics were used: the average completion time of the task, calculated as the total time to move from A to B to C; the average precision, calculated using the MIR250 odometry and markers placed on the ground; and the time spent by the human operator in contact with the RELAX mobile cobot body, which was used to quantify the required effort and attention on the part of the operator during the collaborative task. It is clear that the proposed multimodal interface framework has comparable precision to the physical approach, as when greater accuracy is required (i.e., adjusting the final pose of the cobot on the marker), the human operator moved closer to the cobot and activated this modality while at the same time keeping the flexibility of the UWB interface for the rest of the navigation. This resulted in the time spent in contact with the cobot being reduced by around 44%. It is worth noting that the average completion times with the physical interface and the multimodal framework are very similar, as the operator avoids spending time on the precise positioning of the cobot using the UWB signal, which is the main drawback of this standalone interface.

### 5.3. Multimodal Interface Framework for Collaborative Transportation of Long Payloads


**Experimental Setup**


In the last set of experiments, the RELAX cobot with its end-effector was employed to grasp a long payload of about 2 m in length and 3.5 kg in weight during a collaborative transportation task with a human coworker. The mission, as reported in Figure 21, involves moving two of the above-described payloads from a loading zone to an unloading zone.

In this case, the verbal interface is employed to send the RELAX cobot the command to open or close the end-effector. The physical interface is used during the transportation phase to move the RELAX cobot’s base, and the joypad interface is used to return it from the unloading point to the loading one. It is worth noting that the physical interface is exploited when the operator needs to place the RELAX end-effector precisely below the payload for grasping and transport. The same set of parameters used in the second group of experiments was employed in this last validation scenario.


**Results**


Figure 22 shows the results of the last set of experiments. It can be observed that the 2D pose of the RELAX mobile cobot on the map (the top image) moves from the *Loading Zone* to the *Unloading Zone* twice, i.e., once for each of the long payloads to be transported. Moreover, the center image shows the linear (in the X direction) and angular reference velocities for the mobile base generated by the physical and the joypad interfaces, with the latter used only on the way back to the loading zone from the unloading zone. The torques sensed by the RELAX end-effector are visualized in the bottom image, clearly showing the grasping and release phases during transportation from the loading zone to the unloading zone.

A video of the last two experimental trials is available (https://youtu.be/yUaOi5VZhUM).

## 6. Discussion

In this work, we introduce a novel multimodal–multisensor interface framework for a mobile cobot that combines physical interaction, UWB distance measurements, joypad control, verbal interaction, and gesture recognition to create a safe, intuitive, and accurate method for controlling the mobility and manipulation of the robot. The developed multimodal interface was tested on the new RELAX cobot, which represents another contribution of this work. The hardware design and software implementation of the RELAX cobot are described in depth. The motivation for realizing a custom-designed cobot was the need for a robot that could handle high-payload tasks while remaining adaptable and safe for physical interaction, as the cobots available on the market are not able to provide high payload-to-weight ratio performance. Another essential feature of the RELAX mobile cobot is its ease of integration into existing industrial workstations due to its limited mass and dimensions. In the experimental section, we demonstrate the performance of the custom-designed RELAX cobot arm in terms of payload and repeatability, the effectiveness of the proposed unified multimodal interface in terms of execution time and precision in a partially cluttered navigation/mobility scenario for high-payload transportation, and a task involving transportation of a long payload in collaboration with a human co-worker. The scenarios for which the multimodal interface for mobile cobots is intended can be represented by a constrained workspace with static obstacles; to this end, a future direction of our work will focus on carrying out collaborative transportation tasks autonomously while taking into account possible obstacles in the environment. Moreover, a richer GUI and an easier-to-handle device, such as a smartwatch with wireless earbuds, could be provided for human coworkers. Another research topic of interest regards the control of a mobile cobot fleet with one or more humans to extend the current approach to a multi-robot /multi-human use case. Finally, in terms of evaluation, a set of usability trials will be conducted with a representative sample of users to provide insight into their thought processes and decision-making. After these trials, user feedback will be collected through questionnaires and surveys and analyzed in order to identify any usability or acceptability issues. These might include confusion about performing certain tasks, difficulty in finding certain features or information, frustration with the provided human–robot interface, etc.

## 7. Patents

A patent proposal for the methodology presented in this paper has been filed with application ID IT 102023000016302.

## Figures and Tables

**Figure 1 sensors-23-07735-f001:**
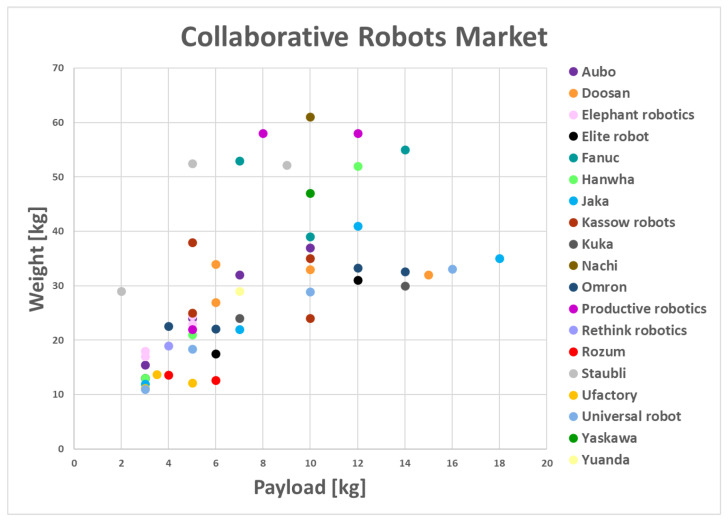
Collaborative robots currently on the market, showing a comparison of weight vs. payload. Copyright [10].

**Figure 2 sensors-23-07735-f002:**
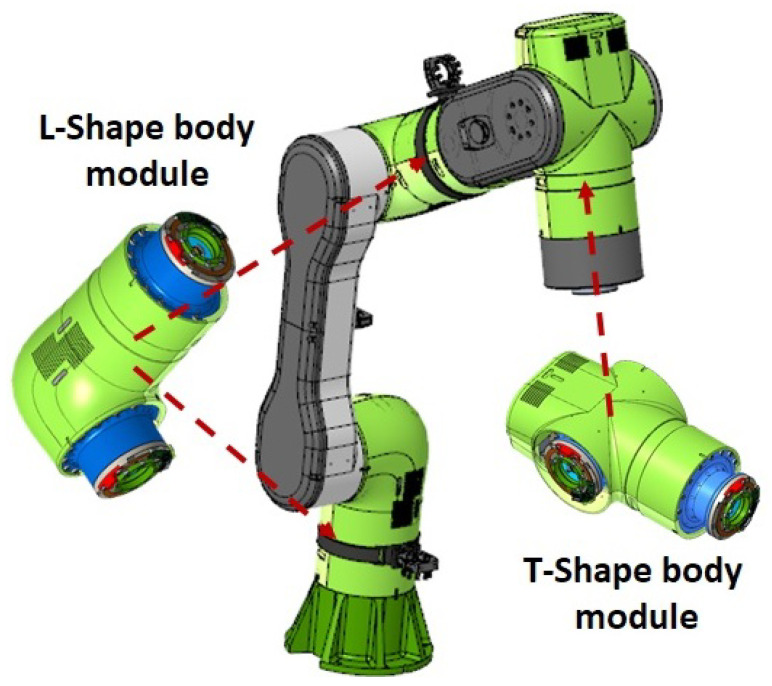
The RELAX modular robot arm is composed of a series of L- and T-shaped 2-DOF modules.

**Figure 3 sensors-23-07735-f003:**
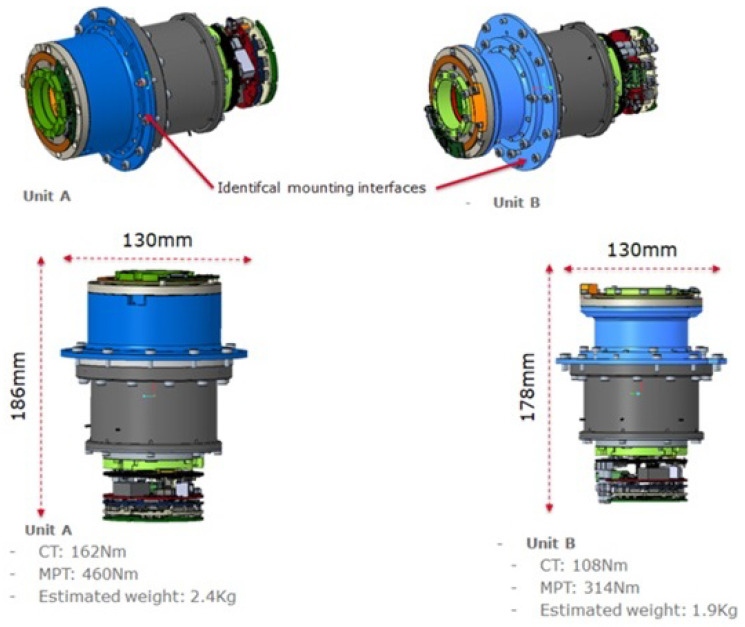
The two custom design actuation units of the RELAX robot arm.

**Figure 4 sensors-23-07735-f004:**
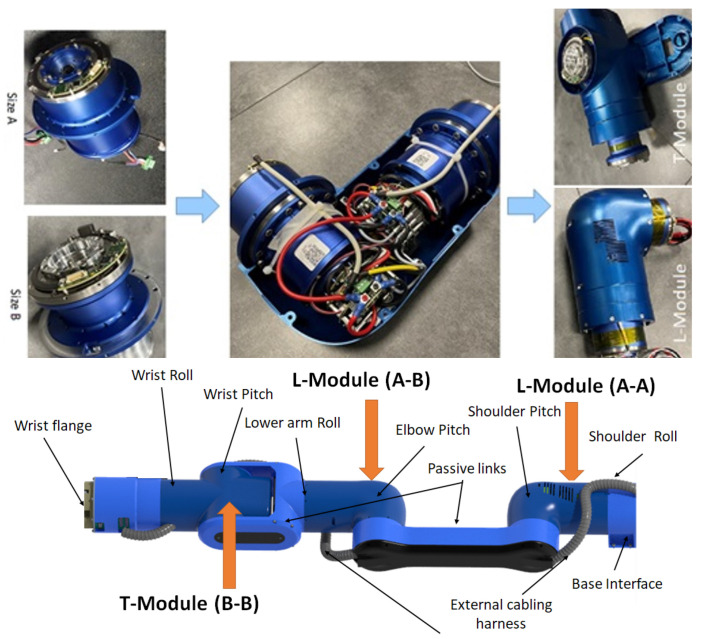
Prototypes of RELAX robot actuators and 2-DOF robot bodies.

**Figure 5 sensors-23-07735-f005:**
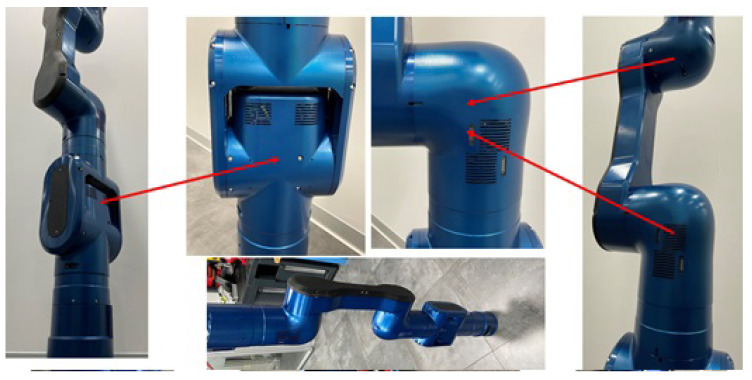
The RELAX high-payload collaborative robot prototype realized with the combination of T- and L-shaped body modules.

**Figure 6 sensors-23-07735-f006:**
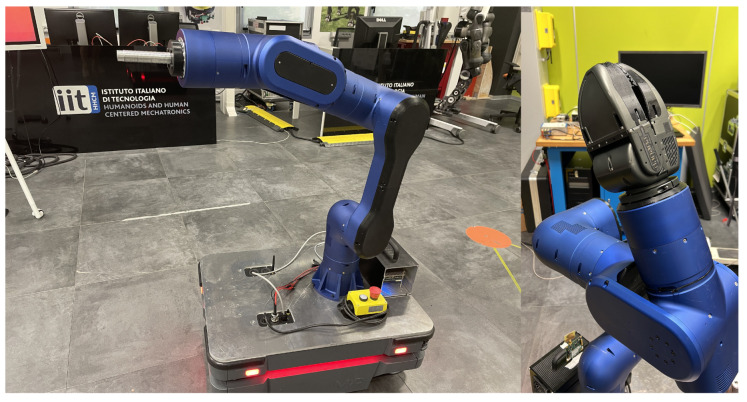
The RELAX high-payload collaborative robot arm integrated with the MIR250. On the left, the configuration used for the first two sets of experiments; on the right, the integration with the generic jaw-type gripper employed in the last sets of experiments.

**Figure 7 sensors-23-07735-f007:**
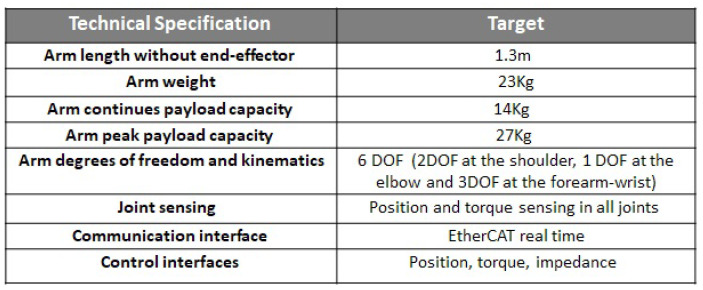
The main specifications of the RELAX high-payload cobot arm.

**Figure 8 sensors-23-07735-f008:**
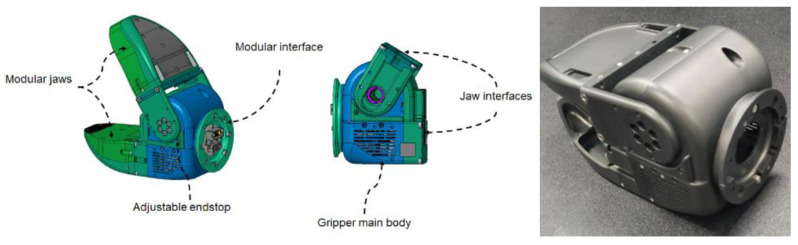
The RELAX generic jaw-type gripper, showing its main body, modular electromechanical interface, and modular jaws.

**Figure 9 sensors-23-07735-f009:**
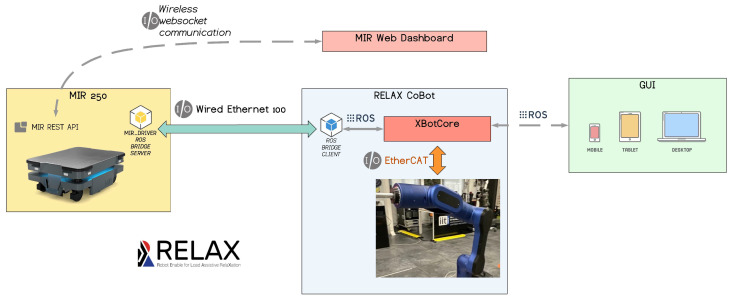
The RELAX mobile cobot software architecture, showing the main components of the MIR250 and RELAX subsystems along with their interconnections.

**Figure 10 sensors-23-07735-f010:**
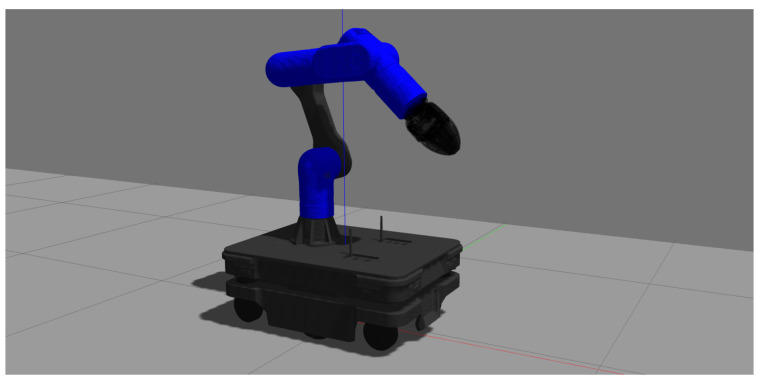
The RELAX mobile cobot as simulated in Gazebo.

**Figure 11 sensors-23-07735-f011:**
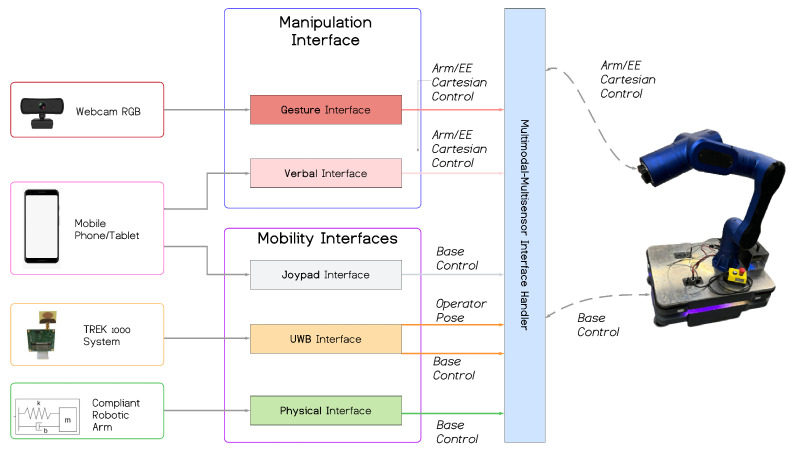
An overview of the proposed unified multimodal–multisensor framework for mobile cobots with the RELAX use case.

**Figure 12 sensors-23-07735-f012:**
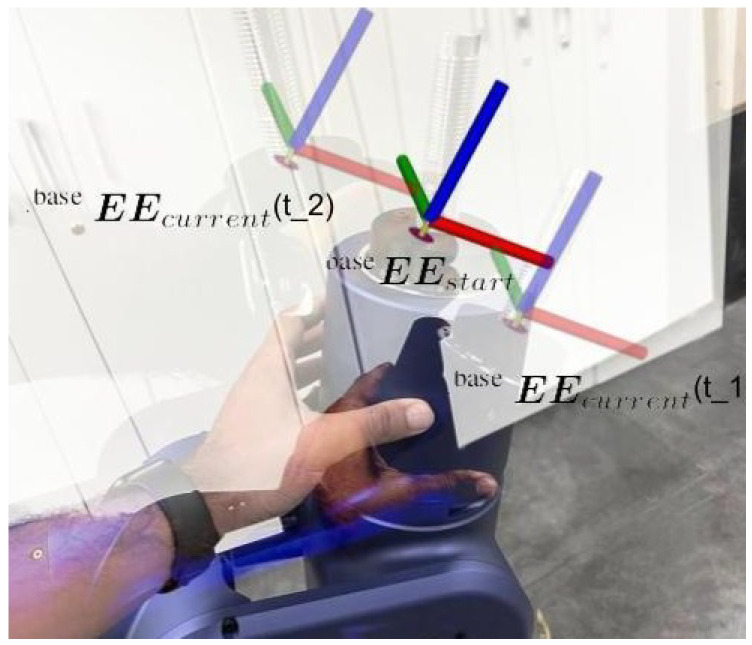
The proposed physical interface, showing the operator interacting with the cobot and generating an end-effector pose displacement thanks to the low impedance setting of the compliant RELAX cobot. This is translated into velocity references for the mobile base, as explained in Equation (Equation 1).

**Figure 13 sensors-23-07735-f013:**
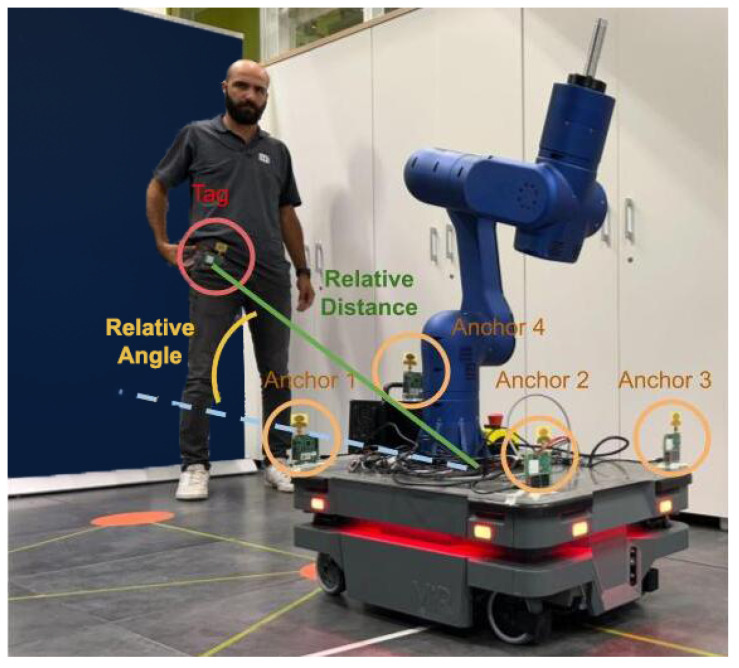
The RELAX mobile cobot introduced in this manuscript was used to validate the proposed unified multimodal–multisensor interface framework. The operator is wearing a UWB tag unit, while a set of UWB anchor units on the cobot are used to compute the relative distance and angle between the operator and the mobile cobot.

**Figure 14 sensors-23-07735-f014:**
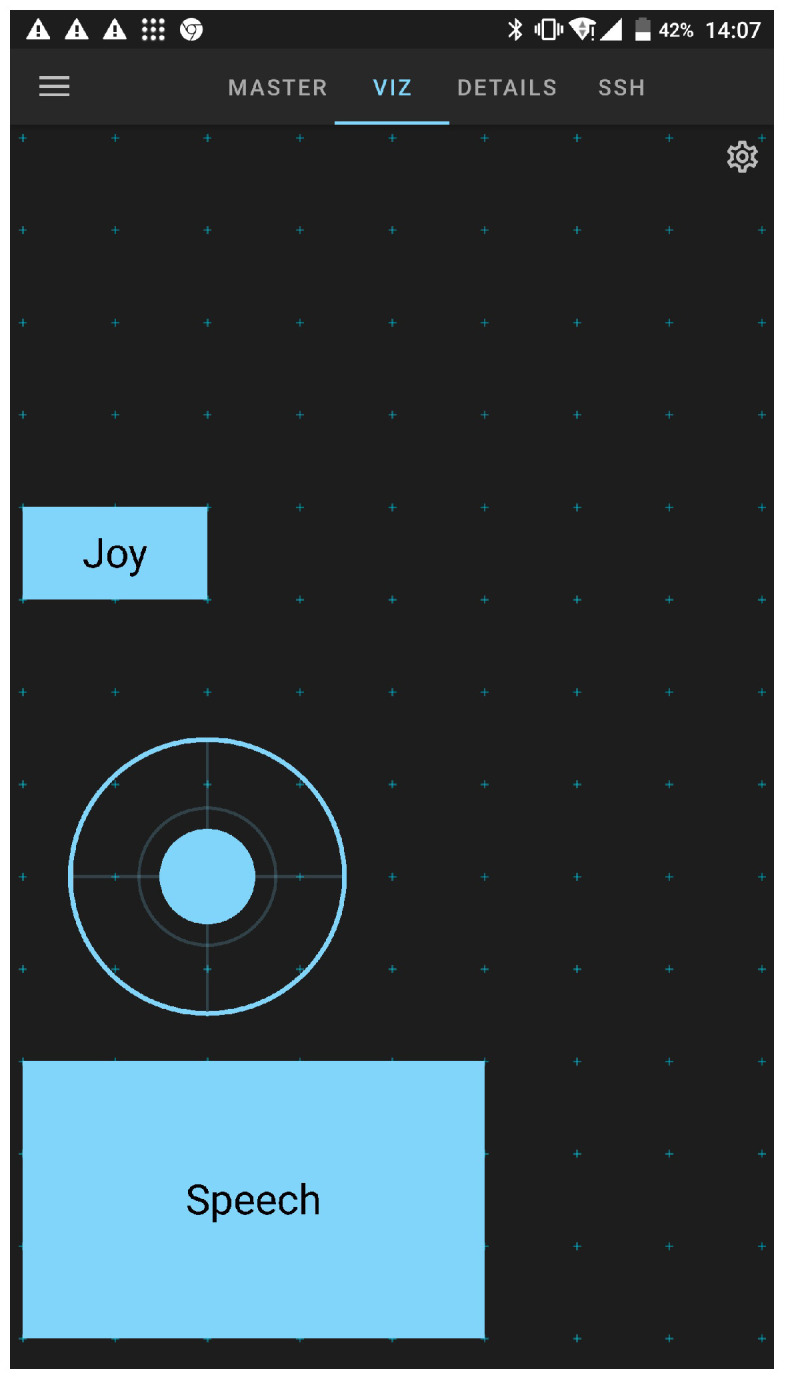
A screenshot of implemented the ROS-Mobile Android app extension, which provides joypad and verbal interfaces with the proposed multimodal framework.

**Figure 15 sensors-23-07735-f015:**
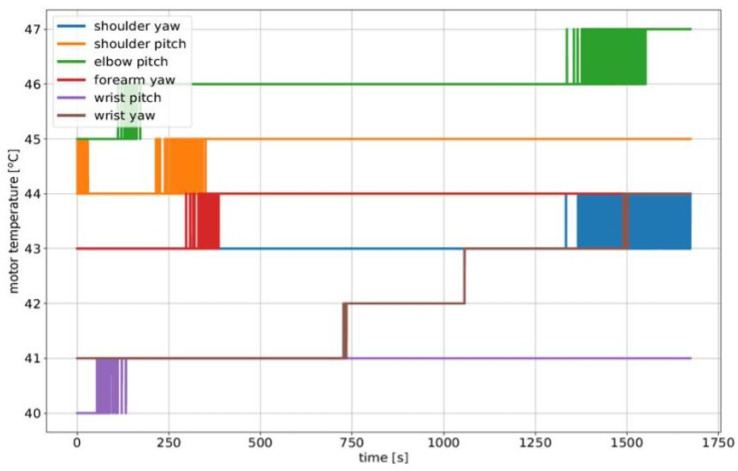
Validation of the RELAX collaborative arm with a 12 kg payload. **Top**: the temperatures of the joints during the 30 min stress test. **Bottom**: the results of the repeatability test.

**Figure 16 sensors-23-07735-f016:**
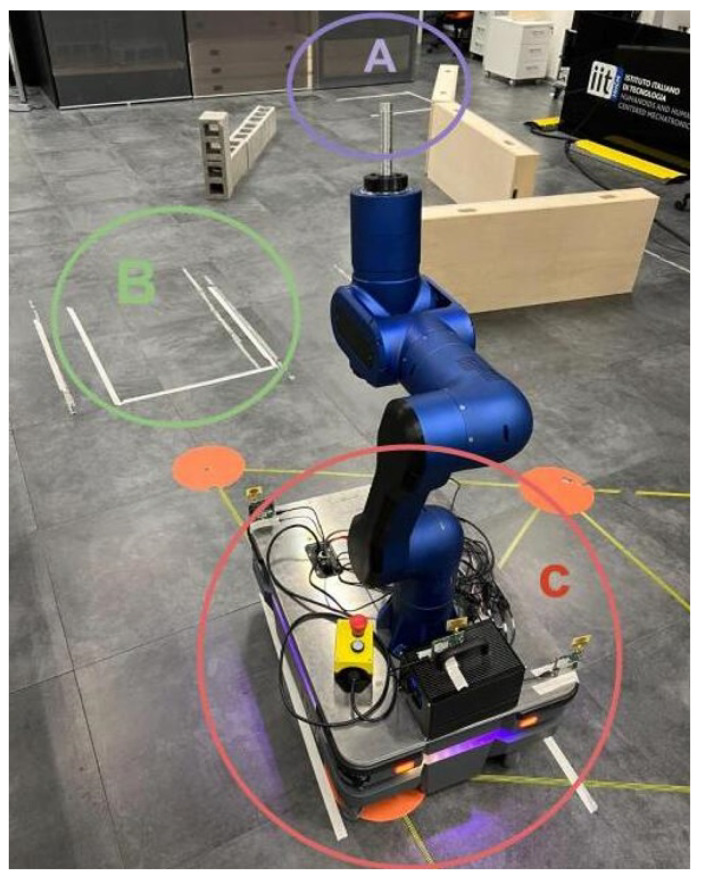
Second experimental validation scenario for the proposed framework. The RELAX mobile cobot must reach points A, B, and C in sequence.

**Figure 17 sensors-23-07735-f017:**
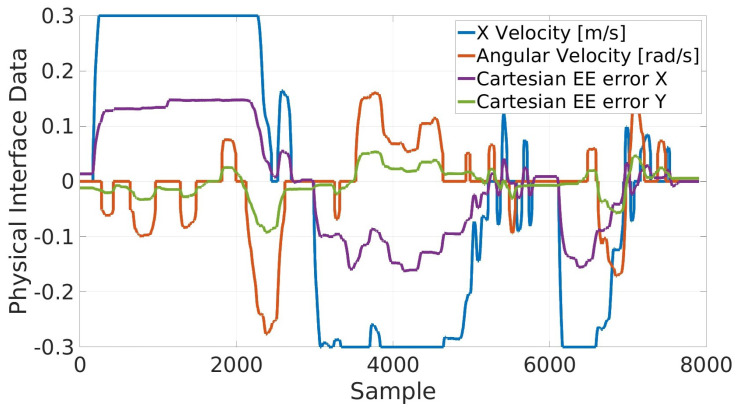
Physical interface results in terms of the commanded velocities for the base and relationship with the Cartesian end-effector computed displacement. It is worth noting that the commanded velocities cannot exceed the user-defined maximum limit, e.g., ±0.3 m/s for the X velocity.

**Figure 18 sensors-23-07735-f018:**
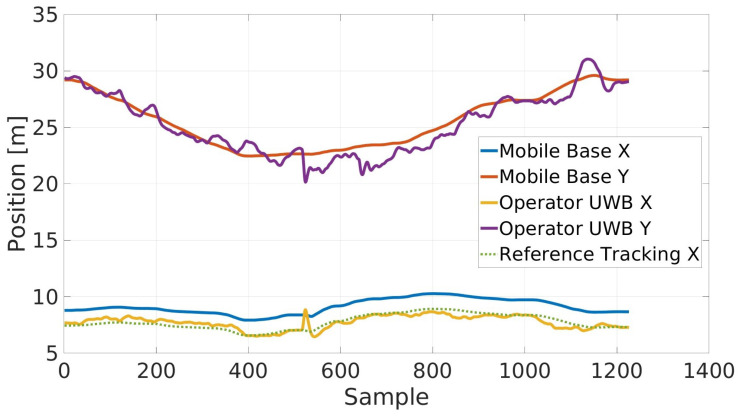
The UWB interface results in terms of the mobile base X and Y position compared with the estimated X and Y position of the UWB Tag (i.e., the operator) used to command the base. The reference tracking line is only shown for the X component, as the Y corresponds corresponds to the Mobile Base Y line.

**Figure 19 sensors-23-07735-f019:**
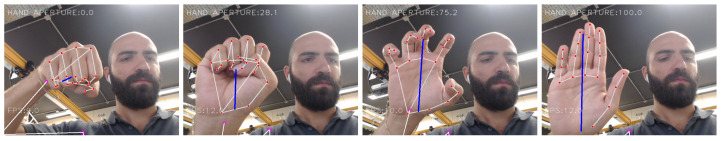
Recognition of the four hand gestures based on the estimated hand aperture.

**Figure 20 sensors-23-07735-f020:**
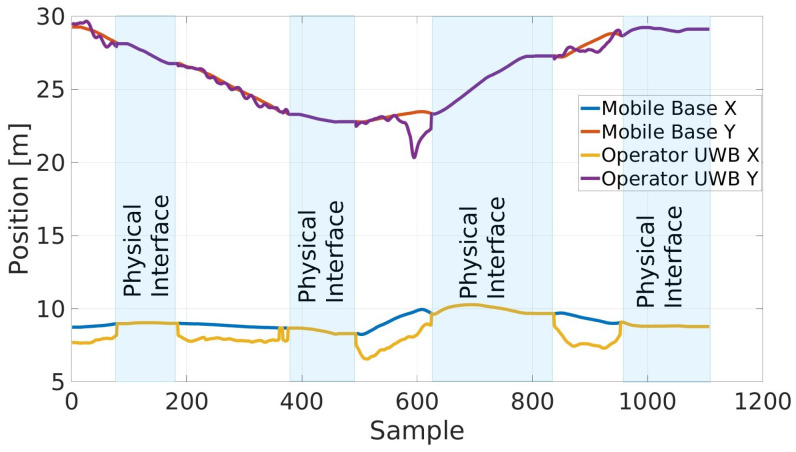
Multimodal interface framework results: the activation of the physical interface is highlighted in light blue, while in the white parts the RELAX mobile cobot is commanded using the UWB interface.

**Figure 21 sensors-23-07735-f021:**
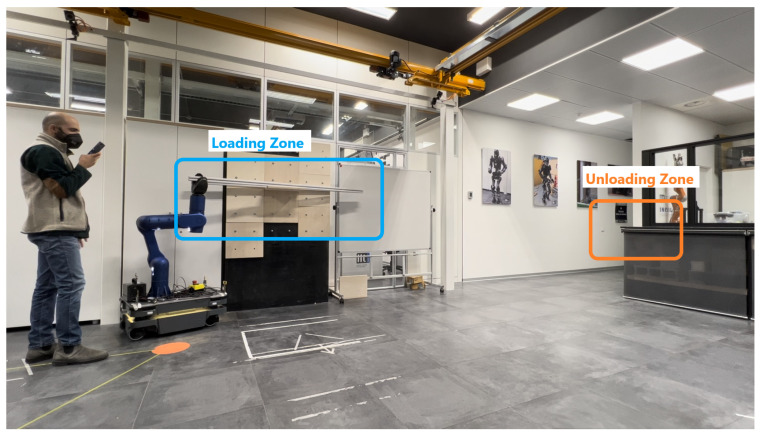
The collaborative transportation task involving the RELAX mobile cobot equipped with end-effector and a human coworker with a mobile phone. The mission involves transporting two long payloads from the loading zone to the unloading zone.

**Figure 22 sensors-23-07735-f022:**
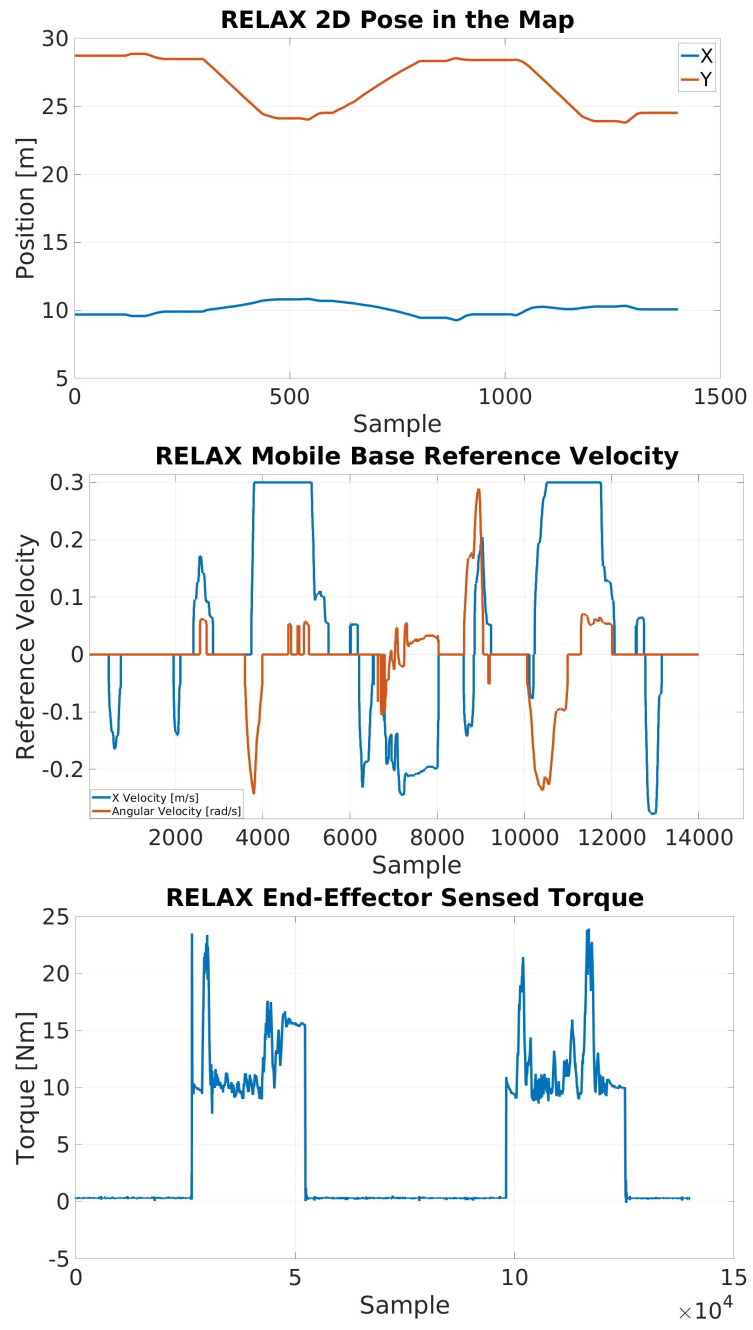
Experimental results from the long payload collaborative transportation task trial. **Top**: the RELAX 2D pose on the map. Middle: the commanded mobile base velocities. **Bottom**: the torque sensed at the end-effector actuator, indicating the grasping and release intervals.

**Table 1 sensors-23-07735-t001:** Features and pros/cons of the presented interfaces.

Interface	Mode	Interaction Range	Accuracy	Signal Quality
**Physical**	Force	Small	High	High
**UWB**	Distance	Medium	Medium	Low
**Joypad**	Velocity	High	Medium	High
**Verbal**	Speech	High	Medium	Medium
**Gesture**	Visual	High	Low	Medium

**Table 2 sensors-23-07735-t002:** Results of the comparison among the proposed mobility interfaces and the multimodal framework.

Interface	Average Completion Time [s] (With Standard Deviation)	Average Precision [m] (With Standard Deviation)	Time in Contact with the Cobot [%]
**Physical**	84±4.8	0.042±0.01	100%
**UWB**	141.6±6	0.073±0.01	0%
**Multimodal**	106.8±9.4	0.049±0.01	56.4%±4.2%

## Data Availability

Not applicable.

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
