# Peer review of "A Unified Multimodal Interface for the RELAX High-Payload Collaborative Robot"

_sensors, 2023, doi:10.3390/s23187735_

Round 1

Reviewer 1 Report

The paper is presented very well. I could observe significant improvement compared to the paper submitted at the conference. The paper requires only a few modifications in the introduction section. The authors are suggested to improve the literature study by referring to the recent cobots. In what way is your cobot different from the existing one? Summarize the findings from experiments under different environments. The video was clear, and we could observe the real performance of the cobot. What are the future extensions? 

Author Response

Dear Editor,
First of all, we would like to thank You and the Reviewers for the careful review of our paper and for your constructive suggestions.
We have done our best to take into account all the Reviewers’ comments duly. We believe that our manuscript has substantially improved after taking into account your valuable insights.
In the attached document, Reviewers’ comments (RC) are reported for convenience, followed by a reply by the Authors (AR). Any modification to the text is highlighted with the color blue in the revised manuscript.

Best regards,
Luca Muratore on behalf of the Authors

Reviewer 2 Report

A novel unified multimodal interface is proposed for facilitating collaboration tasks between humans and mobile collaborative robots. A set of human-robot cooperation tasks are carried out to demonstrate the effectiveness of the proposed multimodal framework. Some suggestions are listed as follows:

In this paper, the problem that the author intends to solve should be further condensed. Particularly, abstracts contain too much detail to highlight the main contributions of the paper.

In the paper, the authors propose an interface framework in which the reader may not care about the specifics of mobile robots. Specifically, Section 2 can be shortened or even deleted.

In the experiments, the authors are advised to supplement the criteria for selecting the parameters. Additionally, the experiment of RELAX high-payload and resilience validation is carried out to demonstrate the RELAX cobot’s high-payload capacity and resilience, is it necessary? As far as I know, this may be determined by the robot’s performance.

Author Response

(The authors gave the same response as above.)

Reviewer 3 Report

The presented work involves introducing a novel unified multi-modal interface that facilitates collaboration tasks between humans and mobile collaborative robots. I believe that the presented work is a significant one, however, there are many issues that must be taken into consideration, as follows: 

1. Missing citation in the first paragraph of the Introduction Section (line 22).

2. The motivation of the presented work was not discussed in the Introduction section.

3. I suggest adding a new section called Related Works, where authors need to add and cite the recent developed relevant systems. 

4. The Introduction Section is not well organized. Authors are required to restructure the Introduction Section. 

5. In Section 2, authors revealed that they have employed the ROS platform. However, authors didn't mention anything about the employed ROS packages nor the developed ROS nodes and services. 

6. A long paragraph exists in Section 3.6 (lines 248-270). I suggest dividing this paragraph into two or more paragraphs. 

7. I believe that authors need to employ more evaluation matrix in order to present the effectiveness of the proposed model. 

8. I suggest adding a new section named as Results, and should show and discuss the obtained results. 

9. I also suggest adding a new section called Discussion, as the presented Section (5. Discussion and Future work) is limited and requires more discussion and analysis. 

Author Response

(The authors gave the same response as above.)

Round 2

Reviewer 2 Report

no other question

Reviewer 3 Report

Authors have addressed all the issues.